# Diagnosis and Management of NREM Sleep Parasomnias in Children and Adults

**DOI:** 10.3390/diagnostics13071261

**Published:** 2023-03-27

**Authors:** Greta Mainieri, Giuseppe Loddo, Federica Provini, Lino Nobili, Mauro Manconi, Anna Castelnovo

**Affiliations:** 1Department of Biomedical and NeuroMotor Sciences, University of Bologna, 40139 Bologna, Italy; 2Department of Primary Care, Azienda AUSL di Bologna, 40100 Bologna, Italy; 3IRCCS Istituto delle Scienze Neurologiche di Bologna, 40139 Bologna, Italy; 4Unit of Child Neuropsychiatry, IRCCS Istituto G. Gaslini, 16147 Genoa, Italy; 5Department of Neuroscience–Rehabilitation–Ophthalmology–Genetics–Maternal and Child Health, DINOGMI, Università degli Studi di Genova, 16132 Genoa, Italy; 6Sleep Medicine Unit, Neurocenter of Southern Switzerland, Ospedale Civico, 6900 Lugano, Switzerland; 7Faculty of Biomedical Sciences, Università Della Svizzera Italiana, 6900 Lugano, Switzerland; 8Department of Neurology, University Hospital, Inselspital, 3010 Bern, Switzerland; 9University Hospital of Psychiatry and Psychotherapy, University of Bern, 3000 Bern, Switzerland

**Keywords:** NREM sleep parasomnias, disorders of arousal, sleepwalking, confusional arousal, sleep terror, sleep-related eating disorder, sexsomnia

## Abstract

Non-rapid eye movement (NREM) sleep parasomnias are recurrent abnormal behaviors emerging as incomplete arousals out of NREM sleep. Mounting evidence on NREM sleep parasomnias calls for an update of clinical and therapeutical strategies. In the current review, we summarize the state of the art and provide the necessary background to stimulate a critical revision of diagnostic criteria of disorders of arousal (DoA), the most common NREM sleep parasomnia. In particular, we highlight the poor sensitivity of the diagnostic items related to amnesia and absence of conscious experiences during DoA episodes, encourage the role of video-polysomnography and home-video recordings in the diagnostic and treatment work-up, and suggest three levels of diagnostic certainty based on clinical and objective findings. Furthermore, we highlight current gaps of knowledge that prevent the definition of standard guidelines and future research avenues.

## 1. Introduction

Non-rapid eye movement (NREM) sleep parasomnias are recurrent abnormal behaviors emerging as partial arousals out of NREM sleep [1,2,3]. While traditionally considered childhood developmental disorders, NREM sleep parasomnias tend to persist more frequently than expected during adulthood [4,5], with a lifetime prevalence of about 7% [6]. Especially during adulthood, they could have variable consequences ranging from physical trauma, diurnal sleepiness [7,8], and psychological distress to legal issues due to violent/aggressive behaviors [9].

The diagnosis of NREM sleep parasomnias is currently performed on the basis of internationally recognized clinical criteria [1,10], meaning the International Classification of Sleep Disorders, Third Edition (ICSD-3) and the Diagnostic and Statistical Manual for Mental Disorders, Fifth Edition (DSM-5) (see Table 1). NREM sleep parasomnias mainly include confusional arousals (CA), sleep terrors (ST), and sleepwalking (SW), three clinical entities lumped together under the name of disorders of arousal (DoA). ST are episodes of abrupt terror associated with strong signs of autonomic arousal; CA are episodes of mental/behavioral disorientation and confusion that occur while the patient is in bed; and SW consists of episodes of ambulation or other complex actions performed out of bed. In addition, two other specific phenotypes—sleep-related abnormal sexual behaviors or sexsomnia and sleep-related eating disorder (SRED)—have over the years gained their own individual attention and nosography. Sexsomnia or sleep sex, characterized by abnormal sleep-related sexual hetero or auto directed behaviors, is currently considered a special sub-type of CA by the ICSD-3 [1] and as a SW sub-type by the DSM-5 [10]. SRED is characterized by recurrent episodes of dysfunctional eating occurring after an arousal during the main sleep period, and is currently considered a NREM sleep parasomnia, independent from DoA by the ICSD-3 and as a SW subtype by the DSM-5 [10]. Our knowledge about sexsomnia and SRED is currently limited; moreover, several questions remain open. Conversely, mounting evidence is available regarding DoA clinical, phenomenological, and physiological correlates [2,4].

The primary aim of this review was to describe the diagnostic evidence available for DoA and to summarize current pharmacological and non-pharmacological therapeutical approaches. Given the more limited information about sexsomnia and SRED, these two entities will be discussed separately throughout this manuscript. The secondary aim was to identify currently available elements in the literature that could improve the diagnostic process, including possible changes in current diagnostic criteria. The third aim was to highlight current gaps in our knowledge that prevent the definition of standard guidelines in order to pave the way to an international diagnostic and treatment consensus.

## 2. Diagnosis of DoA

### 2.1. Clinical Interview

An accurate clinical interview is theoretically sufficient to confirm a diagnosis of DoA according to standard international criteria [1,10] (see Table 1). However, limiting the diagnosis to clinical criteria may present limitations related to the age of occurrence of the episodes and the possibility to collect an accurate description of the episodes from the patients and their relatives/bed partners. 

The presence of recurrent incomplete awakenings (criteria A, ICSD-3) associated with an inappropriate or absent responsiveness to the external environment (criteria B in ICSD-3) represents the very core of the definition of DoA [1]. During CA, patients may sit up in bed, stare straight ahead or look around the room with a perplexed or blunt expression, may vocalize intelligible or unintelligible words, and may often return back to sleep soon after. ST usually begins abruptly, with an intense autonomic activation (tachycardia, tachypnea, diaphoresis, mydriasis). Patients may rise screaming and appear scared and unresponsive to external attempts to interact or calm them down. During SW, patients leap out of bed and wander about the room or the house, or more rarely, may even walk outside. SW patients may perform routine activities like dressing, washing, preparing for school, urinating (sometimes at inappropriate locations), but can also engage in more dangerous actions like moving the furniture, manipulating sharp objects, or jumping out a window [1,11,12,13,14,15]. Prolonged cases associated with sleep driving have been described [16]. Notably, the arousal threshold is higher than wakefulness, and pain processing seems to be altered [17], further exposing these patients to potential dangers. 

The ICSD-3 recognizes some other common features in DoA, meaning a limited or absent cognition or dream imagery during the episodes (criteria C in ICSD-3 and criteria B in DSM-5) and partial-to-complete amnesia for the episodes (criteria D in ICSD-3 and C in DSM-5) [1]. Frequent amnesia significantly undermines history taking and makes critical the presence of caregivers or bedpartners who witnessed the episodes during the clinical interview. Interestingly, the view of DoA as automatic behaviors performed in the absence of any conscious experience has been significantly challenged in the last two decades. In fact, different retrospective studies have shown various degrees of mental/oneiric recall associated with DoA episodes (above 70% of patients across their life span) [18,19,20]. Mental experiences during DoA episodes seem not only limited to single visual scenes or simple mental fragments, and even complex, vivid and long dream-like/hallucinatory experiences may not rule out a diagnosis of DoA [21]. In the only case series where patients with ST were interviewed immediately after their episodes, mental imagery was present in ~ 60% of the cases [22]. The degree of recall may depend on several factors, including the time distance between the clinical interview and the episode’s occurrence, as well as the individual mnesic ability (usually more developed in adults). Of note, the recall rate seems to differ between children and adults, in line with the hypotheses of more immature sleep mechanisms in children (and lower dream recall rate), and with an easier arousability in adults (that may facilitate the incorporation of the dream experience in long-term memory) [5,20]. Thus, we suggest that criteria C and D should be considered optional supportive criteria (especially in children) but not mandatory criteria (see Table 2). Of note, in adults, mental content may also have a discriminatory value against RBD. Indeed, while negative emotions and violent behaviors are as frequent in DoA as in RBD [19], high levels of apprehension associated with misfortune (meaning any threat over which the subject does not have control) is more strictly associated with DoA than RBD. Self-defense or fleeing behaviors are more typical of DoA and direct fighting/aggressive behaviors (where the patient acts the first move) or aggressions by either humans or animals are more frequent in RBD [18,19]. Moreover, the occurrence in at-home settings (and not in a virtual dreaming scenario as in RBD) seems to be very specific for DoA [23].

CA, ST, and SW should be interpreted as a diagnostic continuum on the basis of clinical, epidemiological, physiological, and genetic evidence [2]. Indeed, patients usually display more than one clinical subtype (CA, ST or SW) throughout their lifespan and may even switch from one manifestation to the other within the same episode. Of note, the presence in the same subject of different semiological manifestations across episodes and a lack of stereotypy within the episodes, together with partial interactions with the surroundings (e.g., environment exploration and apparently purposeful object manipulation) seem to be specific for DoA. These elements support the differential diagnosis with sleep-related hyper-motor epilepsy (SHE)—the condition that poses more challenging issues in terms of differential diagnosis, especially in the presence of short episodes [11,24,25]. Contrary to REM sleep behavior disorder (RBD), eyes during DoA episodes are wide open [26], even during minor episodes, and movements more often involve the trunk [27].

The timing and the evolution of episodes are other essential features to consider. Indeed, DoA episodes more frequently—although not exclusively—occur during the first third of the night (when slow wave sleep (SWS) predominates), while RBD more typically occurs in the second half of the night (when REM sleep predominates); SHE tends to manifest more equally across the entire night [28]. Last but not least, DoA episodes usually occur in clusters across the life span, with a frequency of episodes that fluctuates over time, possibly in relation to sleep deprivation and stress [12].

Another fundamental part in DoA clinical history is the identification of predisposing, priming, and precipitating factors like the familiar history [29], the presence of symptoms or signs suggestive for other sleep disorders known to fragment sleep (like sleep apneas or periodic limb movements), an unusual sleep environment, sleep deprivation, sensory stimuli, or medications [30,31].

### 2.2. Questionnaires

Few questionnaires have been validated for screening patients with a clinical suspect of NREM sleep parasomnias and/or quantify their severity. These scales have been validated mainly in adult populations. 

The Munich Parasomnia Screening (MUPS) questionnaire [32] is a self-rating instrument composed of 21 items and provides a quick overview on the occurrence and frequency of parasomnias and other nocturnal behaviors in adults. However, the protocol of validation did not include a video-polysomnography (VPSG) to confirm the diagnosis, as this questionnaire was meant more for screening than for diagnostic purposes [32,33].

The Paris Arousal Disorders Severity Scale (PADSS) [34] is a useful instrument for monitoring DoA clinical symptoms and severity. It is composed by a self-rated scale assessing the occurrence of 17 parasomnia behaviors, their frequency, and consequences. Of note, neither MUPS or PADSS were validated against patients with SHE [35]. 

To this extent, the Frontal Lobe Epilepsy and Parasomnias (FLEP) Scale was developed to differentiate SHE from parasomnias, especially DoA [36]. However, not all the patients in this study had a diagnosis confirmed by VPSG. A successive Italian validation of the same scale [37], in which the diagnoses were confirmed by in-lab VPSG, did not confirm these encouraging results. The scale gave an incorrect diagnosis in 5.6% of cases, which included patients with SHE seizures characterized by nocturnal wandering (mistaken for SW), while around one-third of cases had uncertain diagnostic indications, due to items associated with a risk of misdiagnosis (mainly “recall” and “clustering” of the events, increasing the chance of mistaking RBD for seizures) [37].

The Arousal Disorders Questionnaire (ADQ) was recently validated to screen adults with DoA according to ICSD-3 criteria [38]. The questionnaire includes two parts: the first part is specific for each subtype (CA, ST, SW), while the second part assesses general DoA criteria. This questionnaire has a good interobserver reliability [39], an acceptable sensitivity and a high specificity for the diagnosis of DoA. Excluding the items regarding consciousness and episode recall (ICSD-3 criteria C and D, see Table 3), the sensitivity increased without major changes in specificity, highlighting that these criteria might not be always appropriate for adult cases of DoA. 

Overall, the use of validated questionnaires might support the screening of DoA in non-specialized settings, improve DoA diagnostic accuracy in clinical and research sleep settings, assess the severity of DoA over time, before and after treatment, and guide further diagnostic assessments in case of atypical scenarios or comorbidities.

**Table 3 diagnostics-13-01261-t003:** Scales available for NREM sleep parasomnias screening, diagnosis, or follow-up.

	Validation Sample	Sensitivity	Specificity	Utility
MUPS [32]	65 adult psychiatric patients 50 adult patients with sleep disorders 65 adult HC	ST: 100% CA: 100% SW: 83%	ST: 89% CA: 97% SW: 100%	Screening of parasomnias, sleep-related movement disorders or normal variants
PADSS [34]	73 adult and adolescent active SW/ST 45 adult and adolescent patients with sleep disorders (26 former DoA, 19 RBD) 53 adult and adolescent HC	DoA: 84%	DoA: 88%	DoA severity and follow-up
FLEP [36]	62 adult and children/adolescent patients (31 with SHE, 29 with DoA, 2 with RBD)	SHE: 100%	SHE: 90%	Differentiate SHE from parasomnias
FLEP [37]	71 adult patients (11 DoA, 14 SHE and 46 RBD)	SHE: 71%	SHE: 100%	Differentiate SHE from parasomnias
ADQ [38]	47 adult and adolescent DoA 103 adult and adolescent patients with sleep disorders (56 RBD, 39 SHE, 6 NES, 2 with drug-induced DoA)	DoA: 72% (83% without criteria C and D)	DoA: 96% (93% without criteria C and D)	DoA diagnosis

DoA, Disorders of Arousal; RBD, REM behavior Disorder; SHE, Sleep-related hypermotor epilepsy; NES, Night Eating Syndrome; ST, Sleep Terror; CA, Confusional Arousal; SW, Sleepwalking; HC, healthy control subjects.

### 2.3. Video-Polysomnography

VPSG is usually performed when evaluating patients with atypical NREM sleep parasomnias, meaning cases with red flags, such as (1) adult onset; (2) stereotypical, repetitive, abnormal unnatural postures, ballistic movements or focal motor patterns; (3) high episode frequency; (4) episodes of brief duration; (5) episodes recurring in any part of the night; (6) other suspected sleep disorders; and (7) presence of dissociative symptoms [40] during daytime/other major psychiatric comorbidities [41] (see Figure 1).

VPSG assumes an important role in the differential diagnosis between DoA and SHE [35], RBD or parasomnia overlap disorder [42], as well as in sleep-related (psychogenic) dissociative disorders [43]. Furthermore, VPSG traces can give important information regarding other potential sleep disorders triggering DoA, such as obstructive sleep apneas and periodic limb movements (PLM) (up to 33% of the cases in adults) [44].

Moreover, in the assessment of DoA, video synchronized with EEG adds critical information. The probability of catching a typical episode during VPSG is approximately 30–60% [44,45,46,47]. This percentage largely depends on the frequency/severity of parasomnia episodes, the life period when the recording is performed, the patient’s age, and the sensibility of the scorer to minor/brief nocturnal episodes [44,45,46,47], which represent the majority of episodes captured in-laboratory.

#### 2.3.1. Video Data

The systematic assessment of DoA semiology has been performed in few studies [11,12,26,48,49,50,51,52], but a conclusive agreement is still lacking (Table 4).

The first attempt to categorize DoA behaviors was performed by Kavey et al., who described three patterns: (1) dramatic episodes with patients moving abruptly and impulsively in bed; (2) episodes characterized by sitting or kneeling on the bed; and (3) episodes with kicking or gesticulation [50].

Later, in a study demonstrating the value of sleep deprivation as a diagnostic tool in adult SW, Joncas classified DoA behaviors on a three-point Likert scale: (1) simple behaviors in bed; (2) complex behaviors in bed; and (3) complex behaviors outside the bed [51].

Derry et al. proposed another classification according to the behavioral and emotional patterns observed during the episodes, identifying three main behavioral patterns: (1) arousal behaviors; (2) non-agitated motor behaviors; and 3) distressed emotional behaviors [11] (see Table 4). The following elements were considered as strongly suggestive for DoA: yawning, scratching and prominent nose-rubbing, rolling over in bed, sobbing/sad emotional behavior, physical or verbal interaction, internal or external triggers (noise, cough, snore), waxing and waning pattern, indistinct offset, no full awakening after events with complex behaviors, prolonged duration (>2 min), discordance between severity, and duration of reported event and recorded event. Conversely, features, such as sitting, standing, or walking, preceding “normal” arousal, brief arousals (up to 10 s) without definite semiological features of epilepsy, or fearful emotional behavior did not discriminate between SHE and DoA.

A more recent study on 45 patients (334 DoA events) proposed a different hierarchical classification of increasing motor complexity [12,52]: (1) simple arousal movements (SAMs); (2) rising arousal movements (RAMs or pattern II); and (3) complex arousal movements (CAMs or pattern III) (see Table 4). All patterns can be accompanied by eyes opening, vocalizations, screaming, and various levels of interaction with the external environment. SAMs are minor episodes most frequently recorded during standard VPSG, hence the recognition of these episodes could be useful in establishing a definitive DoA diagnosis in patients with a clear clinical history of DoA but without the VPSG documentation of a “major” episode [52]. Furthermore, these simpler episodes were recently compared to seizures with paroxysmal arousals, meaning brief initial fragments of a major SHE seizure [24]. The characterization of their duration (a median of 12 s versus 5), sleep stage at onset (NREM stage N3 vs N1/N2), movement progression (waxing and waning versus continuous), and associated behaviors, such as exploratory behaviors, are important to differentiate DoA from SHE [24].

Eye opening is another essential feature, as demonstrated in a sample of 52 adult DoA patients (see Table 5) [26]. However, this criterion was only validated against controls and does not rule out a diagnosis of SHE (that also may start with abrupt eye opening [24]).

All in all, the identification of distinctive motor signatures in sleep disorders requires larger studies as well as comparison with physiological movements and arousals of healthy sleepers, analytically characterized only in a few studies so far [53,54]. Besides, the improvement of technologies, e.g., motion tracking devices, might serve to quantify movement variables such as speed, which are currently entirely dependent on subjective observer evaluation.

#### 2.3.2. EEG Data

Overall, DoA episodes tend to occur out of SWS (80%) in the first third of the night [28]. However, independently from the time of occurrence during the night, the occurrence of at least one major event outside N3 is highly specific for SHE, while the occurrence of at least one minor event during N3 is suggestive for DoA [28].

DoA post-arousal patterns have been divided into three different categories according to the observed EEG frequencies (slow, mixed, and fast) [55], with a predominance of slow arousals in DoA patients compared to normal control subjects. DoA episodes are immediately preceded by a diffuse increase in EEG slow wave activity (SWA) [56] and a suppression of sigma power [57,58]. This pattern does not seem to differ between simpler or complex DoA behaviors [57], and qualitatively resembles the pattern observed prior to physiological arousals—although perhaps richer in SWA and poorer of higher (beta) activity [59]. Delta activity could be visually detected in almost half of the behavioral episodes from SWS and in a minority (~20%) of episodes from stage 2 [60]. Rhythmic and synchronous delta activity is preferentially associated with relatively simple and less complex behavioral episodes during SWS [60]. Of note, delta activity increase seems to be inhomogeneous across the brain [58], and to coexist with an increase of higher frequencies in other brain regions [61] or even in the same brain region [57,58,62,63]. SWA connectivity also drastically changes prior and during DoA episodes [58,64]. Overall, the aforementioned quantitative EEG analysis of DoA episodes, while offering a deeper understating of DoA pathogenesis [2,65], has not been tested against potentially difficult differential diagnoses, such as SHE. Moreover, EEG signal analysis requires complex analyses that prevent its immediate use in clinical practice at the current stage.

The search for sleep trait abnormalities independent from clinical episodes has assumed particular relevance [55], as a single overnight in-laboratory VPSG is often negative for full-blown DoA episodes, or may allow the detection of only minor episodes undistinguishable from brief SHE episodes. Early studies disclosed an increase in SWS sleep [66]. However, this finding was not confirmed by other studies, which showed a substantially normal sleep macrostructure, comparable to healthy controls [7,18,24,45,49,51,67,68]. Hyper-synchronous delta wave bursts are a relatively common finding, although with relatively low specificity [69]. Slow wave activity topographical distribution seems to be altered in patients compared to controls [70]. SWS temporal dynamics are also impaired, with an abnormal homeostatic build-up across the night [71] and a selective increase in the number of arousals/awakenings during SWS, not paralleled by a similar fragmentation in other sleep stages [26,45,66,71,72]. The N3 distribution index, defined as ([number of epochs of N3 in the first half of SPT – number of epochs of N3 in the second half of the SPT]/total number of N3 epochs), specifically captures SWS homeostasis [28]. This index is higher in healthy control subjects, intermediate in patients with DoA and lower in patients with SHE. However, specific cut-offs, or sensitivity/specificity values, are not currently available for this index. SWS instability [3,31] recently demonstrated potential diagnostic validity through the definition of two indexes (the SWS fragmentation index and the slow/mixed arousal index) in both children and adults [45,72]. The SWS fragmentation index is essentially the number of AASM [73] arousals, awakenings, and slow wave bursts > 3 s associated with an increase in chin EMG - per hour of SWS. The slow/mixed arousal index is the sum of the mixed (high and low EEG frequencies) and slow (low frequencies) arousal indexes in SWS [45]. Strikingly, the slow arousal pattern somewhat corresponds to the A1 phase of the so-called cyclic alternating pattern (CAP) [74], an indirect measure of the arousal level fluctuations, which is known to be increased in DoA, especially in the first two sleep cycles [49,75,76].

In order to be included in standard clinical practice, these indexes need to be validated versus other sleep pathological conditions, such as sleep breathing disorders or SHE, where sleep instability and CAP rate has been proven to be even higher than DoA [77].

**Table 5 diagnostics-13-01261-t005:** Indexes proposed on the base of video-EEG.

Episode-Independent VPSG Indexes				
	Cut-Off	Sensitivity	Specificity	Predictive of
SWS fragmentation index [45,72]	>6.8/h (adults) >4.1/h (children)	79% 65%	82% 84%	DoA vs HC
Slow/mixed arousal index [45,72]	>2.5/h (adults) >6/h (adults) >3.8/h (children)	94% 60% 69%	68% 100% 82%	DoA vs HC
Episode-Dependent VPSG Indexes				
Eye opening during SWS interruptions [26]	≥2	94%	77%	DoA vs HC
Major event outside N3 [28]	≥1	79%	95%	SHE vs DoA
Minor event during N3 [28]	≥1	73%	72%	DoA vs SHE

DoA: disorders of arousal; HC: healthy controls; SHE: sleep-related hypermotor epilepsy.

### 2.4. Home-Video-Recordings

Despite being the current gold standard for the assessment of complex nocturnal sleep behavior, supervised VPSG is expensive, time consuming, and often implies long waiting lists. Furthermore, as highlighted in the previous section, the probability of catching at least one parasomnia episode during a single-night VPSG assessment is far from being satisfactory. This percentage largely depends on the frequency/severity of parasomnia episodes, the patient’s age, and the sensibility of the scorer to minor/brief nocturnal episodes [44,45,46,47], which represent the majority of episodes captured in-laboratory. Indeed, the differential diagnosis of brief nocturnal episodes by both video and EEG is often extremely difficult [11]. In this perspective, the analysis of homemade video recordings of nocturnal episodes may be an important supportive diagnostic tool for DoA [78]. It offers multiple advantages: wide availability, low-costs, the possibility of recording patients in their usual sleep environment and of repeated recordings, increasing the chances of capturing multiple and possibly longer and more complex episodes compared to the standard in-laboratory setting.

Even in the case of partial recordings by parents/bed partners, when the onset of the episode is missed, video-recordings could be a useful tool for the differential diagnosis of nocturnal events, as it is not the onset of the episodes but the evolution and the offset of the events that has a discrimination value between DoA and SHE [78]. Furthermore, repeated home videos using professional high-quality infra-red motion-detector cameras can now record and store consecutive episodes from several nights.

Up to now, 2 single case reports highlighted the usefulness of monitoring nocturnal behaviors in adults with NREM sleep parasomnias with an infrared camera in the home environment for 5 and 2 weeks, respectively [21,79]. More recently, a study of 20 adults with frequent DoA episodes, recorded for at least 5 consecutive nights, confirmed a good feasibility, acceptability, and clinical value of home-video recordings in DoA [80]. An average of at least 3 nights was required to capture at least 1 event and events were usually more complex than those recorded during VPSG.

### 2.5. Differential Diagnoses

DoA must be differentiated by a number of other sleep disorders [81] (see Table 6): REM-related parasomnias, like nightmare disorder, RBD, and parasomnia overlap disorder [82]. SHE and sleep related psychogenic dissociative disorder (SRDD) are probably the trickiest conditions that might simulate DoA [40,43], in the absence of objective findings. Nocturnal panic attacks may also simulate ST [83,84,85]. Medication- or substance-induced nocturnal confusion must be always ruled out before considering a DoA diagnosis.

### 2.6. Therapy

Current evidence on both pharmacological and non-pharmacological interventions in NREM sleep parasomnias is hampered by: (1) the relative low number of patients that reach specialistic attention; (2) the lack of agreement between specialized sleep centers; (3) the lack of animal models and of a clear understanding of neural circuits implied in NREM sleep parasomnias; and (4) the fact that treatment efficacy is particularly insidious to test, given the frequent episode-related amnesia and the consequent relatively low reliability of subjective reports. Only one study using an objective outcome (number of episodes captured with home-video recordings) has been conducted to date. This study suggested a reduction in the frequency of episodes in eight DoA patients treated with Clonazepam [80]. Treatment strategies currently available for DoA has been extensively reviewed elsewhere [86,87,88,89,90]. We will briefly summarize them below and in Table 7.

#### 2.6.1. First-Level Treatment

DoA treatment remains based on the following general mainstays:(1)reassurance regarding the absence of other associated severe neuropsychiatric conditions and the possible spontaneous resolution with age [15,91];(2)instruction to parents/bed partners to refrain from interacting with/trying to awake the patient (to avoid worsening or lengthening the episodes or violent reactions by patients) [9];(3)education on sleep hygiene (e.g., regular sleep-wake routine, quiet and dark bedroom environment, avoidance of sleep deprivation, or of excessive alcohol intake) [31,92];(4)specific instructions for environmental safety (removal of firearms or other weapons from home, removal of any bedside object or furniture that could cause injuries, installation of bedroom door/window/stairway alarms or locks, sleeping on the ground floor) [93];(5)treatment of sleep comorbidities, especially sleep apneas—even when very mild [94]—and periodic leg movements [95];(6)identification and removal of presumed inducing pharmacological agents (meaning almost all psychotropic medications, as well as many other drug classes, introduced in temporal association with the beginning of DoA symptoms [31]).

#### 2.6.2. Medications

Pharmacologic treatment is generally considered in case of frequent episodes, despite the resolution and removal of all possible priming and precipitating factors for severe episodes (violent or non-violent behaviors, dangerous for the patient or others and/or with potential legal consequences, significant functional impairment like excessive daytime sleepiness, weight gain from nocturnal eating, or personal embarrassment) [90]. Patients or their parents/caregivers should be advised that drugs for DoA are all “off label” and sign a written consent [90].

##### Benzodiazepines and Z-Drugs

While the Z-drug Zolpidem has a relatively high risk of inducing a NREM parasomnia episode [6,96,97], especially in predisposed subjects, low to moderate doses of benzodiazepines (BDZ)—and especially of the long-acting Clonazepam (0.25–2 mg)—have proven to be effective in treating DoA [86,90]. While smaller clinical case series reported mixed results [94,98], larger studies showed stable and rather convincing results. In particular, in a case series conducted in 1989 on 28 adult patients affected by SW and ST, ~84% had a rapid and sustained response to Clonazepam [99]. Subsequently, a larger case series published in 1996 on 58 patients with SW and ST demonstrated the efficacy, dose stability, safety, and low abuse potential of chronic, nightly, low/moderate doses of Clonazepam [100]. The efficacy of Clonazepam was confirmed in 2012 by a retrospective study on 57 subjects affected by DoA, with a response rate of about 75% [101]. A similar success rate (~72%) has been reported in 2019 by a larger retrospective study where 173 out of 350 NREM sleep parasomnia patients were exposed to BDZ, primarily Clonazepam [92]. The same study revealed a lower efficacy of Z-drugs, antidepressants, and melatonin (49%, 37%, and 38%, respectively) [92]. The efficacy of Clonazepam has also been occasionally described in some cases of DoA induced by neuroleptics [102] and in cases of sleep-driving and violent behaviors [16,100]. More anecdotical and sometimes contradictory results are available for other BDZ, like Alprazolam [100,101], Triazolam [103], Temazepam [101], Flurazepam [50], Clorazepate [104], and Diazepam [104,105,106].

##### Antidepressants

The literature on DoA and antidepressants (AD) remains controversial [107]. DoA, and especially ST, may respond to low doses of AD such as Clomipramine [101], Imipramine [101,108], Trazodone [109], Sertraline [101], and Paroxetine [110,111]. Fewer and sometimes contradictory cases have been reported regarding Paroxetine in SW, with Paroxetine potentially resolving [111,112] or inducing [113] SW. The appearance of SW has also been associated with other AD, like Bupropion [114,115], Mirtazapine [116], and Reboxetine [117]. As mentioned above, in a large retrospective study including 54 NREM sleep parasomnia patients treated with various AD (32% with Fluoxetine; 20% with Citalopram; 8% with Mirtazapine and Trimipramine; 7% with Amitriptyline, 5% with Paroxetine, Sertraline, Clomipramine, and Trazodone; and 5% with Imipramine and Venlafaxine) the success rate was below 40%.

##### L-OH-Tryptophan

In children with ST, L-OH-tryptophan (a precursor of serotonin) at high doses (2 mg/Kg) is probably the medication of choice [118], as suggested by an open clinical trial conducted in 45 children with ST (31 treated vs. 14 untreated). This study reported 94% efficacy (meaning >50% reduction in the number of reported episodes) at the 1-month follow-up and 84% efficacy at the 6-month follow-up, after a single period of treatment over 20 consecutive days.

##### Melatonin and Melatonin Agonists

A relatively recent retrospective study showed the potential use of extended-release Melatonin in 37 adult patients with NREM sleep parasomnias and in 18 patients with parasomnia overlap disorder, with an effect rate of 38% [92]. A surprisingly higher response rate (88%) was reported by another recent smaller retrospective study (8 patients treated with Melatonin—although the formulation was not specified) [98]. Melatonin has been anecdotally reported to be effective in younger children (1 case in a 36-month-old child with ST [119]), including syndromic cases (1 child affected by Asperger syndrome and 1 child with a chronic sleep-phase onset delay [120]).

Only one case report is available on Ramelteon, a Melatonin receptor agonist, which resulted effectively in a boy with attention-deficit/hyperactivity disorder (ADHD) and ST/SW [121].

##### Other Medications

Other medications have been rarely reported in the treatment of DoA. Curiously, osmotic release oral system methylphenidate (OROS-MPH) was anecdotally reported to be effective in two adult SW cases with no comorbid ADHD [122]. In clinical practice, antiepileptics/stabilizers like Gabapentin, Levetiracetam, or Carbamazepine are sometimes used, despite the lack of scientific data, with mixed results.

##### Non-Pharmacological Interventions

Psychotherapy is free from any side effect and may represent an alternative approach for the treatment of DoA, particularly important in children and pregnant women.

Psychotherapy may work by reducing priming and triggering factors of DoA, such as sleep deprivation and other unhelpful behaviors (like alcohol or Z-drug consumption), as well as stress and anxiety. Indeed, stress [12,123,124], emotional/behavioral problems, and personality traits have been linked to NREM sleep parasomnias [124,125,126,127,128], although the specific pathway and direction of this association remains unclear. In selected cases, the treatment of specific underlying psychopathology may improve parasomnia symptoms [129], although this possibility requires further confirmation. Lastly, some affected patients may feel ashamed and have negative self-perceptions due to involuntary acts during parasomnia episodes. Psychotherapy may help them increasing the awareness for their dysfunctional beliefs and in managing the emotional distress/conflicts over their own judgments [130].

##### Scheduled Awakenings

When the occurrence of DoA episodes is highly predictable, arousing the patient 15–20 min prior to the usual onset of a typical episode for 1 to 4 weeks may induce sustained positive results, at least in children with SW [131], ST [132], or CA [133,134]. However, it may be highly demanding for parents and it cannot be applied when there is no specific timing for the episodes.

##### Hypnosis

Curiously, clinical hypnosis is the best documented non-pharmacological option for DoA [135,136,137]. Hypnotherapy received specific attention since the earliest definition of DoA, with case reports and small case series on SW in adults [138,139], ST or other complex dangerous behaviors in children [140,141,142,143], and more recently, even in a case of childhood parasomnia overlap disorder (POD) [144].

The efficacy of hypnotherapy was also observed in 2 larger case series of 27 [145] and 17 patients with SW and ST [146] that used a short hypnotherapy protocol (1–6 sessions and 1–2 sessions, respectively). While the first study reported an effect rate of 75%, the second one reported an efficacy of 50% at 18 months, and of 65% at a 5-year follow-up in SW (*n* = 11) and 20–25% at both 1.5-month and 5-year follow-up in ST (*n* = 6). These improvement rates cannot be univocally ascribed to hypnotherapy due to the possibility of spontaneous recoveries and/or improvements due to other subsequent treatments.

One single-blind (rater-blind), modified crossover design study evaluating a brief protocol (6 sessions) of specialized hypnotherapy for severe SW, free from other psychiatric illness, revealed a long-lasting improvement in both subjective and objective symptoms at 1-year follow-up [147].

##### Other Specific Protocols

The efficacy of other relaxation techniques such as autogenic training and the psychodynamic approach to emotional conflicts (related to parasomnia behaviors themselves or external factors, considered as triggers of parasomnia episodes) have been sporadically reported [148,149].

In 2019, a retrospective study described the positive effect of specific non-pharmacological interventions such as mindfulness-based stress reduction (MBSR; *n* = 7), cognitive behavioral therapy (CBT) for stress and anxiety (CBTs-a; *n* = 10), and CBT for insomnia (CBTi; *n* = 13) in patients with NREM sleep parasomnias free from any other sleep comorbidity [92]. The success rate was of 80% (32 out of 40). The selection of these specific interventions was based on the individual clinical presentation (presence of significant stress and anxiety levels or of insomnia symptoms).

##### Integrated Psychotherapy Protocols

In 2013, a large retrospective study on 103 patients described 6 patients effectively treated with non-pharmacological procedures that included sleep hygiene and behavioral intervention, deep relaxation, self-hypnosis, and rudiments of cognitive therapy [101].

A case series published in 2021 on three subjects suggested that the flexible integration of multiple interventions (tailored to individual cases) is effective for treating DoA. Interventions ranged from education about parasomnias and their precipitating factors, mitigation of environmental triggers to stress management, cognitive-behavioral interventions focused on parasomnia-related anxiety, and hypnosis [150].

The same year, a study on 46 NREM sleep parasomnia patients assessed the effect of a 5-session program (CBT-NREMP) that covered psychoeducation on DoA precipitating and priming factors; sleep hygiene and sleep behavioral techniques (sleep rescheduling to optimize homeostatic regulation, stimulus control to re-establish an association between the bed/bedroom and sleep); body-based relaxation techniques (mindfulness-based body scan, progressive muscle relaxation); and cognitive strategies to manage anxiety at night. The authors found a significant, although small, effect on the severity (but not the frequency) of parasomnia symptoms, sleep quality and anxiety/depressive symptoms [151]. More recently, a randomized controlled trial on a CBT protocol specifically developed for parasomnias (CBTp) produced statistically significant improvements in parasomnia frequency, severity, as well as in sleep continuity [152].

## 3. SRED

The first clinical series of SRED were described by the psychiatrist Carlos Schenck in 1991 (19 cases) and then 1993 (38 cases), who reported patients attending a sleep clinic for a combination of sleep and eating disorders [153,154]. Patients were mostly women (66%), with a mean age of 39 years, and half of them (44%) were overweight due to night-eating.

SRED is currently defined by ICSD-3 as the presence of recurrent episodes of dysfunctional eating that occur after an arousal during the main sleep period, plus the presence of at least one of the following features: consumption of peculiar forms or combinations of food or inedible or toxic substances; sleep-related injurious or potentially injurious behaviors performed while searching for food or while cooking food; and adverse health consequences from recurrent nocturnal eating. Patients must also display partial or complete loss of conscious awareness during their eating episodes, with subsequent morning amnesia. The disturbance must not be better explained by another sleep, psychiatric or medical disorder, medication, or substance use [1].

The relationship between SRED and DoA is suggested by the fact that SRED patients display: (1) a high frequency of past or current SW, which is the predominant sleep disorder associated with night-eating [155]; (2) eating episodes during the first half of the night; (3) several arousals from SWS sleep [156]; (4) the association with conditions provoking sleep fragmentation such as restless legs syndrome (RLS), PLM and sleep apneas [153,154,157,158,159], along with recurring chewing and swallowing movements during sleep [160], as well as with drugs [161]; and (5) “out of control” eating episodes without any feeling of hunger or thirst, and with the possible ingestion of editable, unpalatable, or even poisonous substances, such as cigarettes, cleaning products, or animal food [156,162,163].

However, none of the eating episodes in medication-free patients captured by VPSG case series happened during intermediate states of sleep and wakefulness, typical of DoA. Patients usually awoke from NREM sleep (both N2 and N3 stage) and started eating within 1 min from the awakening [156].

Historically, SRED has been linked to so-called night eating syndrome (NES). NES was initially described in 1955 by eating disorder specialists in 20 adults with treatment-resistant obesity, morning anorexia (skipping breakfast), hyperphagia in the evening/night (eating more than half of daily calories after 7 PM), and insomnia [164]. The principal distinction between SRED and NES is the presence of awareness during the eating episodes with subsequent recall of the episodes [156]. Indeed, SRED patients are thought to eat in a “half sleep, half awake” state with no morning recollection of their eating episodes, while NES patients typically eat in a state a full or almost full awareness [165]. However, many SRED patients can display various degrees of awareness, even during the same night [165], challenging the idea of a clear-cut distinction between SRED and NES. Furthermore, several studies reported overlapping clinical features between SRED and NES, including nightly binging at multiple nocturnal awakenings, a female prevalence, a chronic course, a primary morbidity of overweight and sleep impairment, the response to similar treatments, a familial recurrence and the frequent co-existence of mood oscillations [165]. In addition, a VPSG case series on patients that satisfied clinical criteria of NES, described a nocturnal ingestion sub-type with a VPSG profile similar to SRED, with episodes occurring within 1 min from a NREM sleep awakening [156]. This led many authors to consider SRED and NES under the framework of a spectrum of nocturnal eating behaviors characterized by variable degrees of consciousness.

The knowledge on SRED therapy is limited, due to the paucity of randomized controlled trials, the high drop-rate and the small sample size of previous observational studies [162,163]. The treatment of patients with SRED should be tailored on each specific situation. Any drug causing SRED must be withdrawn. Comorbid sleep disorders addressed and treated. Among pharmacological options [154,162,166], Topiramate [163], dopaminergic agents [167], Clonazepam [168], and selective serotonin reuptake inhibitors (SSRI) such as Sertraline [169,170] should be considered.

## 4. Sexsomnia

Sexsomnia is currently considered a CA subtype and includes a wide range of hetero or self-directed amnestic sexual behaviors, ranging from seeking the bedpartner, masturbation, complete sexual intercourse or spontaneous orgasms [171]. Patients are usually amnestic. Their bedpartners report asexual attitude either similar to their habitual one or different (more gentle or more aggressive) [171]. The prevalence of the disorder is unknown and likely underestimated, since patients may be unaware of their disorder and/or may be psychologically ashamed to the point of avoiding medical help [172]. Up to now, only a few case series have been reported [171,172,173,174]. From the available literature, a male-prevalence emerges, although this data may be biased by the fact that male patients are at higher risk of forensic implications. The age range is wide [171,175,176]. Comorbid sleep conditions that may act as triggers for sexsomnia episodes include obstructive sleep apneas, PLM, sleep bruxism, or even sleep-related head jerks [173,174,177,178].

VPSG data showed a similar sleep profile to DoA, with a higher rate of SWS awakenings than healthy matched controls, often with dissociated EEG features (mixed slow and fast EEG rhythms) [171]. Episodes have been captured in eight VPSG cases [174], consisting in six males and mostly episodes of sleep masturbation (seven out of eight patients). Only one out of eight cases had full sexual intercourse [174]. In addition to these cases, a penile erection occurrence during SWS has been recorded [171].

No systematic pharmacological trials have been performed. From anecdotical cases, Clonazepam, Lamotrigine, Trimipramine, Fluoxetine, Paroxetine, Escitalopram and Duloxetine had positive effects [179]. In some cases, the resolution of comorbid sleep apneas was sufficient to obtain full remission [174,180,181], while other cases required a multi-modal therapeutical approach, including pharmacotherapy, cognitive behavioral therapy, and hypnosis [179].

## 5. Conclusions and Future Directions

On the basis of the reviewed literature, we believe that the time is now ready to update current DoA diagnostic criteria.

First, as highlighted in Section 2.1, there is now compelling evidence that ICSD-3 criteria C and D regarding amnesia and mental content do not fully fit many adult DoA cases and a minority of children cases. Thus, they should be considered only as supportive and not as mandatory criteria. 

Second, contrary to RBD and many other sleep disorders, objective sleep data are not currently included in DoA formal diagnostic criteria, despite an increasing amount of literature on VPSG in DoA (highlighted in Section 2.3). Maintaining this perspective may possibly have detrimental effects on patients’ clinical management and limit the consistency and reliability of research on the topic. Thus, we propose three levels of diagnostic certainty based on clinical, VPSG, and homemade video findings. The diagnosis of DoA should still be possible when only based on the clinical evaluation (in very typical cases and/or where objective recordings could not be performed), probable when typical DoA episodes are documented by homemade videos and/or when the VPSG only shows brief/short behavioral episodes (in the context of supportive clinical data), and definite when the VPSG documents typical full-blown DoA episodes out of NREM sleep. A typical clinical history, along with typical episodes at VPSG and a negative EEG, may by itself exclude a diagnosis of SHE and strongly support a DoA diagnosis. Nonetheless, other differential diagnoses such as SRDD, although rare, may represent a confounding element, especially in adults.

Third, in the presence of a clinical continuum of nocturnal manifestations, it might be useful to add a distress/disability criterion in the ICSD-3 manual, in line with the DSM-5 criterion D (see Table 1). This additional criterion might guide physicians in the clinical management and treatment of patients with DoA. Furthermore, in order to better define when DoA manifestations should deserve clinical attention, future research should address the bulk of the iceberg, meaning all patients that do not currently seek or have access to third level sleep centers. This could be obtained by means of screening tools as those described in Section 2.2. Specific epidemiological surveys are also needed to clarify the real prevalence of sexsomnia and SRED.

Building on these three premises, we herein suggest new DoA diagnostic criteria (Table 2). Although we are aware that a change in international diagnostic criteria will require an extensive expert debate and consensus, we hope that our proposal could inspire future ICSD-3 and/or DSM-5 editions. In the meantime, we stimulate research to address current critical gaps in our scientific knowledge.

In particular, larger studies using a blinded and randomized design—in a heterogeneous population of patients complaining of abnormal nocturnal motor events using a concomitant VPSG recordings—are warranted to support the use of home-video recording in clinical practice. Furthermore, a closer cooperation with technical manufacturing companies will be necessary to develop medical devices specifically designed for this porpoise (Box 1). In particular, it will be fundamental to improve motion tracking, to secure online data storage, to develop a dedicated software for scoring video events, and portable home-EEG devices coupled with infrared video cameras [80]. Body part recognition using multi-stage deep learning may also allow for the automatic evaluation of nocturnal behaviors, with results possibly even superior to human inspection. Machine learning algorithms on EEG signals may also reveal their utility in the next future in the differential diagnosis of minor episodes. Lastly, in order to further support an objective diagnosis even in the absence of major behavioral episodes, the validation of current (Section 2.3) or new objective sleep indexes (like the SWS fragmentation index) should be performed on larger samples and against possible differential diagnoses.

Finally, in contrast with the many efforts devoted to the improvement of the diagnostic process, it should be highlighted that therapeutical strategies for NREM sleep parasomnias are still mainly based on observational studies and case reports (low or very low quality of evidence according to the GRADE system) [182]. Randomized placebo-controlled studies will be necessary to verify current pharmacological and non-pharmacological knowledge (Box 1). In particular, randomized control trials (RCT) should focus on Clonazepam in adults and 5-L-OH Tryptophane in children, the two molecules that displayed higher effectiveness. These studies must also include VPSG and indexes of sleep fragmentation. Once again, the use of home-videos, and possibly, portable home-video-EEG devices, may allow a better evaluation of DoA phenotype, frequency, and severity, a better follow-up of DoA treatment response in an ecological setting. 

Box 1Research agendaEncourage the use of DoA questionnaires into clinical practice and test them in children and into first-level general practitioner contexts.Reinforce the use of home video-recordings into clinical practice and promote the development of screening portable devices incorporating video and at least one EEG channel.Test sensitivity/specificity of existing and new objective VPSG-based indexes on differential diagnoses.Foster multi-center pharmacological (e.g., Clonazepam in adults and L-OH-Tryptophane in children) and non-pharmacological placebo-control treatment studies (e.g., integrated CBT protocols).

## Figures and Tables

**Figure 1 diagnostics-13-01261-f001:**
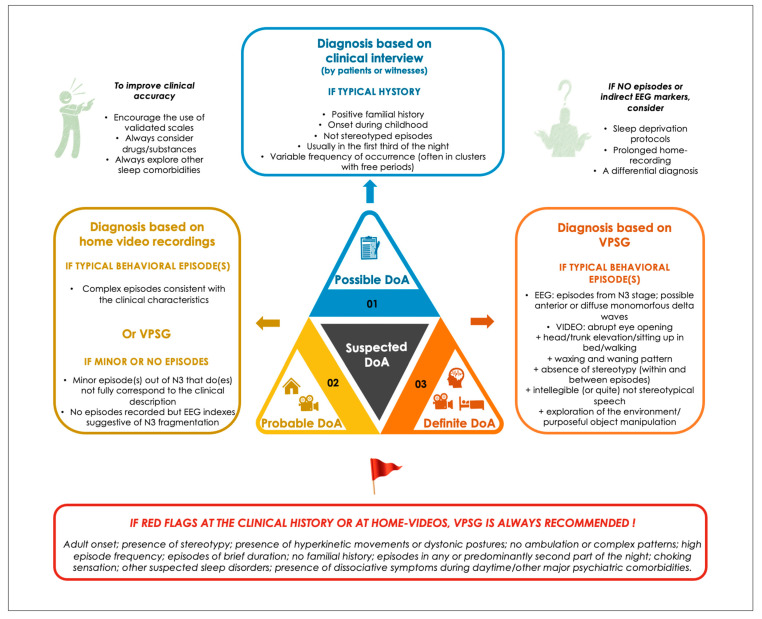
Proposed levels of certainty in DoA diagnosis.

**Table 1 diagnostics-13-01261-t001:** International diagnostic criteria for non-rapid eye movement sleep parasomnias according to ICSD-3 and DSM-5.

	Definition
ICSD-3	
Clinical Manifestation	Recurrent episodes of incomplete awakening from sleep. Inappropriate or absent responsiveness to efforts of others to intervene or redirect the person during the episode. Limited (e.g., a single visual scene) or no associated cognition or dream imagery. Partial or complete amnesia for the episodes Notes: The events usually occur during the first third of the major sleep episode. The individual may continue to appear confused and disoriented for several minutes or longer following the episode.
Exclusion	E. The disturbance is not better explained by another sleep disorder, mental disorder, medical condition, medication, or substance abuse.
Subtypes	Sleep terror. Sleepwalking. Confusional arousal. − Sleep-related abnormal sexual behaviors.
DSM-5	
Clinical Manifestation	Recurrent episodes of incomplete awakening from sleep, usually occurring during the first third of the major sleep episode, accompanied by either one of the following: − Sleepwalking: repeated episodes of rising from bed during sleep and walking about. While sleeping, the individual has a blank, staring face; is relatively unresponsive to the efforts of others to communicate with him or her; and can be awakened only with great difficulty. − Sleep terrors: recurrent episodes of abrupt terror arousals from sleep, usually beginning with panicky scream. There is intense fear and signs of autonomic arousal, such as mydriasis, tachycardia, rapid breathing, and sweating, during each episode. There is relative unresponsiveness to efforts of others to comfort the individual during the episodes. No or little (e.g., only a single visual scene) dream imagery is recalled. Amnesia for the episodes is present.
Distress/Disability	D. The episodes cause clinically significant distress or impairment in social, occupational, or other important areas of functioning.
Exclusion	E. The disturbance is not attributable to the physiological effects of a substance (e.g., a drug of abuse, a medication). F. Coexisting mental and medical disorders do not explain the episodes of sleepwalking or sleep terrors.
Subtypes	Sleep terror type. Sleepwalking type. − with sleep-related eating. − with sleep-related abnormal sexual behaviors.

ICSD-3, International classification of sleep disorders, third edition; DSM-5, Diagnostic and statistical manual for mental disorders, fifth edition.

**Table 2 diagnostics-13-01261-t002:** Suggestions for new DoA diagnostic criteria.

	Definition
Clinical Manifestation	Recurrent episodes of incomplete awakening from sleep. Inappropriate or absent responsiveness to efforts of others to intervene or redirect the person during the episode. Supportive features: The events usually occur during the first third of the major sleep episode. The individual may continue to appear confused and disoriented for several minutes or longer following the episode. Limited (e.g., a single visual scene) or no associated cognition or dream imagery (especially in children). Partial or complete amnesia for the episodes (especially in children).
Distress/Disability	D. The disturbance causes clinically significant distress or impairment in mental, physical, social, occupational, educational, or other important areas of functioning, as indicated by the report of at least one: Daytime sleepiness. Disturbed sleep (of the patient or the bedpartner). Self-injuries. Injuries to others. Psychological issues (e.g., feeling of shame, anxiety, fear to go to bed).
Exclusion	E. The disturbance is not better explained by another sleep disorder, mental disorder, medical condition, medication, or substance abuse.
Levels of certainty	Possible: the diagnosis is based only on clinical interview (witness or self-report description). Probable: the diagnosis is based only on clinical interview (witness or self-report description) and: − Home video recordings document typical behavioral episode(s) consistent with the clinical characteristics retrieved from the patient’s or witness’s interview. − The video-polysomnography documents minor episode(s) that does not fully correspond to the clinical description retrieved from the patient’s or witness’s interview. Definite: video-polysomnography documents typical behavioral episode(s) out of NREM sleep.

**Table 4 diagnostics-13-01261-t004:** VPSG studies assessing DoA semiology.

	Patients	Episodes	Patterns	Against
Blatt et al. [48]	24	NS	Sitting up in bed, performing semi-purposeful movements and gestures and lying down again; raising both legs in the air several times in the supine position; talking or screaming.	HC
Zucconi et al. [49]	21	129	Confusional arousal and abnormal arm or trunk movements with a “tendency” of the patient to get out of bed and possibly to stand up.	HC
Kavey et al. [50]	10	47	(1) Dramatic episodes with patients moving abruptly and impulsively in bed; (2) episodes characterized by sitting or kneeling on the bed; (3) episodes with kicking or gesticulation.	HC
Joncas et al. [51]	10	44	(1) Simple behaviors in bed; (2) complex behaviors in bed; (3) complex behaviors outside the bed.	
Derry et al. [11]	23	57	(1) Arousal behaviors (eye opening, head elevation, staring, face rubbing, yawning, scratching, moaning, and mumbling); (2) non-agitated motor behaviors (sitting forward, manipulation of close objects and searching behaviors); (3) distressed emotional behaviors (screaming with fearful facial expression, frantic searching or evasive behaviors).	SHE
Baldini et al. [12]	45	334	(1) Simple arousal movements characterized by head flexion/extension (pattern IA); head flexion/extension and limb movements (pattern IB), head flexion/extension and partial trunk flexion/extension (pattern IC); (2) rising arousal movements when the patient sits up in bed; (3) complex arousal movements that coincide with SW.	HC
Barros et al. [26]	52	953 N3 interruptions	Eye opening, raising head, exploring the environment, expression of fear/surprise, speaking, raising trunk, interaction with environment, sitting, screaming, standing up/walking.	HC

NS, not specified; HC, healthy controls; SHE: sleep-related hypermotor epilepsy.

**Table 6 diagnostics-13-01261-t006:** DoA differential diagnoses.

	DoA	RBD	SHE	SRDD	Nightmare Disorder	Nocturnal Panic Attacks
Age at onset	Infancy	>50 years, rare in children/adolescent (mostly Narcolepsy patients)	Any age	Variable	Children>adults	Variable—adults
Gender	M = F	M in sleep clinics, equal in general population	M > F	F > M	M = F	F in DP/NP, M in isolated NP
Predisposing or priming conditions	Family history, stressful situations, sleep deprivation	-	-	Traumatic events/abuse/major psychopathology	Traumatic events	Mostly DP attacks
Triggers	+	-	+/−	+	+	+
Occurrence during the night	Usually, first third	Second part of the night	Any time	W close to bedtime	Second part of the night	First third
Sleep stage	N3	REM sleep	N1, N2	Prolonged W	REM sleep	Transition N2–N3
Frequency (number of episodes/ night)	One major, possible multiple minor	Variable, from one to several	Several	From one to several	Usually, one	Usually one, occasionally > one
Episodes duration	1–10 min	Variable, seconds to several minutes	Seconds to 3 min	>1 h	3–30 min	2–8 min
Stereotypy	-	-	+	-	-	-
Mental content	Short scenes involving misfortune or threat	Complex scenarios, variable content from aggression to positive	-	-	Nightmare, intense fear	-
Consciousness at the end of the episodes	Impaired	Preserved	Variable	Preserved	Preserved	Preserved
Recall	Rare in children, > in adults	Frequent	Inconstant	Inconstant	Present	Vivid recall of fearful sensations, not related to dream scenario
Clinical consequences	Sleepiness, injuries, daytime impairment	Injuries	Injuries, daytime impairment	Self-inflicted injuries	Mood disturbance, daytime impairment, bedtime anxiety	Daytime disfunction

M, males; F, females; DoA: disorders of arousal; DP, diurnal panic; NP, nocturnal panic; PTSD, post-traumatic stress disorder; SHE: sleep-related hypermotor epilepsy; RBD: REM-sleep behavior disorder; SRDD: sleep-related dissociative disorder; W, wake; N1: sleep stage 1, N2: sleep stage 2; N3: sleep stage 3.

**Table 7 diagnostics-13-01261-t007:** Treatment approaches available for DoA.

Non-Specific Interventions			
Reassurance			
Parents/bedpartners education			
Sleep hygiene			
Environmental safety			
Treatment of other sleep disorders			
Removal of triggering medications			
**Pharmacological interventions**			
	Age group	Efficacy	Type of evidence
Clonazepam	Adults	>70%	Large case series Case reports/small case series
Other benzodiazepines	Adults	Insufficient/ contradictory results	Case reports/small case series
Antidepressants	Adults	Insufficient/ contradictory results	Case reports/small case series
5-L-OH-Tryptophane	Children	>80%	Open clinical trial (ST only)
Melatonin	Adults - Children	Insufficient/ contradictory results	Small retrospective case series/case reports
**Non-pharmacological interventions**			
	Age group	Efficacy	Type of evidence
Scheduled awakenings	Children	Insufficient results	Small case series
Hypnosis	Adults - Children	Positive results	Rater-blind trial/Case series
Integrated CBT protocols	Adults	Positive results	Retrospective case series Randomized control trial
Other specific non-pharmacological approaches	Adults - Children	Insufficient results	Case reports/Case series

## Data Availability

Not applicable.

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
