# Peer review of "Diagnosis and Management of NREM Sleep Parasomnias in Children and Adults"

_diagnostics, 2023, doi:10.3390/diagnostics13071261_

Round 1

Reviewer 1 Report

I read with great interest the manuscript “Diagnosis and management of NREM sleep parasomnia as in children and adults”. I thought it was a well written review of parasomnias. I was delighted they included the difficulty of differentiating NREM parasomnias from parasomnia mimics, especially sleep related dissociated disorder. The only recommendations I have is adjusting the spacing and design of the tables.   Line 370-372 recommend adding reference that supports you should refrain from trying to awake a patient.

Author Response

We thank the reviewer for his/her appreciation of our work. The tables are dependent on the journal formatting but we have adjusted the spacing and design whenever possible, to ensure more clarity to the reported text. We have added the following reference to line 370-372: Siclari F, Khatami R, Urbaniok F, Nobili L, Mahowald MW, Schenck CH, et al. Violence in sleep. Brain J Neurol 2010;133:3494–509. We thank the reviewer for the oversight of missing a reference.

Reviewer 2 Report

Thank you for the opportunity to review this well-written and engaging paper. This is a comprehensive review of the literature on the management of NREM parasomnias, that will likely be cited by others in the field. I thank the authors for their attention to detail and conscientiousness in preparing this manuscript. The  manuscript is appropriate in length and organization and is properly referenced. The tables and figure are overall helpful and add to the manuscript's value. I may quibble somewhat with the complexity of the figure, but this can be overlooked. 

Author Response

We thank the reviewer for his/her comments and appreciation of our work. We are aware that the figure is a little complex but we have tried to put the important information, necessary to understand each section of the figure. We have now modified the figure by enlarging the content of each section, in order to be more easily readable.

Reviewer 3 Report

"Diagnosis and management of NREM sleep parasomnias in 2 children and adults" is an important paper and should be published.

The language (grammar, orthography, sentences, ...) should be improved.

The authors citated themselve several times.

Author Response

We thank the reviewer for this comment and the appreciation of our work. The manuscript has now been revised by a native English. We are aware that we cited some of our works, but NREM parasomnia is a topic of interest of our different groups (Bologna, Lugano, Genova) and we are working on several different aspects, going from clinic-semeiological characteristics to neurophysiological ones. In addition, we have put over 150 references in the paper and we cited several other authors and groups working throughout the world on the same topic.